# Nanoscopic Characterization of Starch Biofilms Extracted from the Andean Tubers *Ullucus tuberosus*, *Tropaeolum tuberosum*, *Oxalis tuberosa*, and *Solanum tuberosum*

**DOI:** 10.3390/polym14194116

**Published:** 2022-10-01

**Authors:** Cynthia Pico, Jhomara De la Vega, Irvin Tubón, Mirari Arancibia, Santiago Casado

**Affiliations:** 1Food and Biotechnology Science and Engineering Department, Technical University of Ambato, Ambato 180207, Ecuador; 2Isabrubotanik S.A., Ambato 180150, Ecuador

**Keywords:** starch biofilms, Andean tubers, atomic force microscopy, nanoscience, melloco, mashua, oca

## Abstract

The replacement of synthetic polymers by starch biofilms entails a significant potentiality. They are non-toxic materials, biodegradable, and relatively easy to gather from several sources. However, various applications may require physicochemical properties that might prevent the use of some types of starch biofilms. Causes should be explored at the nanoscale. Here we present an atomic force microscopy surface analysis of starch biofilms extracted from the Andean tubers melloco (*Ullucus tuberosus*), mashua (*Tropaeolum tuberosum*), oca (*Oxalis tuberosa*), and potato (*Solanum tuberosum*) and relate the results to the macroscopic effects of moisture content, water activity, total soluble matter, water vapor permeability, elastic properties, opacity and IR absorption. Characterization reveals important differences at the nanoscale between the starch-based biofilms examined. Comparison permitted correlating macroscopic properties observed to the topography and *tapping* phase contrast segregation at the nanoscale. For instance, those samples presenting granular topography and disconnected phases at the nanoscale are associated with less elastic strength and more water molecule affinity. As an application example, we propose using the starch biofilms developed as a matrix to dispose of mouthwash and discover that melloco films are quite appropriate for this purpose.

## 1. Introduction

The production of edible films extracted from a natural and organic source has potential in the food, nutraceutical, pharmaceutical, biomedical, and cosmetic industries [1]. Its green chemistry and reduced environmental impact make biopolymers a suitable alternative to traditional synthetic petroleum-based methods for many applications.

Starch constitutes one of the best-known biofilm bases [2]. This polysaccharide can join numerous glucose molecules by glycosidic bonds, creating a structure that can be used as a multipurpose organic matrix. However, the properties of these films are dependent on their chemical composition. Amylose/amylopectin percentage, plasticizer (as glycerol) concentration, or the presence of other trace elements during sample preparation, such as lipids and proteins, may entail important structural differences in each resulting starch film [3,4,5,6,7,8]. Hence, a proper selection of the starch source is of the utmost importance to optimize the starch-based film attributes for a particular application. 

Tubers are a possible starch source. Although potato is widely used for this purpose, it is not the only tuber containing this carbohydrate. A plethora of other Andean tubers offer emerging and promising alternatives, which also present low oxygen permeability, good transparency, odorlessness, and resistant structural properties. Melloco, ulluco, or olluco (*Ullucus tuberosus C.*), mashua (*Tropaeolum tuberosum R. and P.*), and oca (*Oxalis tuberosa Mol.*) have been proposed as appropriate substitutes [9,10,11,12]. However, starch layers extracted from diverse sources may acquire different properties due to their composition variability [9,13].

Macroscopic dissimilarities among starch films extracted from diverse sources can be detected by comparing the resulting measurements of various layers yielded at equal conditions from different tubers. Nevertheless, exploration of the causes producing this variability requires focusing on the structural conformation of the films. Nanoscopic topographic characterization opens the possibility of visualizing the segregation of material heterogeneities at this scale. Atomic force microscopy (AFM) permits directly measuring the surface of a sample with nanoscopic resolution in ambient conditions [14,15,16,17]. Furthermore, this technique is also able to provide information about energy dissipation differences on the sample at this scale, allowing the detection of valuable structural conformation characteristics.

Here we show a comparative analysis of starch films extracted from melloco, mashua, oca, and potato (*Solanum tuberosum* L.). We measure their optical contrast, colorimetry, opacity, moisture content (MC), water activity (WA), total soluble matter (TSM), water vapor permeability (WVP), infrared optical absorption, and elastic properties under identical circumstances and compare the results to our nanoscopic surface characterizations using AFM. This study provides a double objective: offering data from biofilms obtained from uncommon Ecuadorian starch sources and establishing a relationship between surface attributes at the nanoscale and macroscopic properties. Our comparison proves the connection, showing that an AFM analysis on the surface of the starch-based biofilms is an easy and fast way to select the best kind of starch-based biofilm, depending on the specific demanding application. It is of notable importance in those cases where a complete chemical characterization is not straightforward.

As an application utility, we explore the possibility of adding a mouthwash to these films and subject them to a tasting experiment. Melloco samples were selected as the best valued starch-based film for this purpose. Coincidentally, in dry conditions, these samples display a consistent nanoscopic granular surface topography coupled with a clearer phase segregation contrast, which can be related to better solubility and lower tensile strength. This proves the worth of an AFM inspection to optimize biofilm selection, depending on the application required.

## 2. Materials and Methods

### 2.1. Starch Extraction

Starch was extracted using a modification of the method of Neeraj et al. [18]. As starch sources, melloco (*Ullucus tuberosus C.*), mashua (*Tropaeolum tuberosum R. and P.*), oca (*Oxalis tuberosa Mol.*), and potato (*Solanum tuberosum* L.) were used. Fresh Andean tubers were washed, peeled, diced, and ground. The pulp was blended with 1 L of distilled water for 5–6 min, then filtered using a cheesecloth having a pore diameter of 120 mesh. The filtrate was collected in a glass beaker and allowed to stand for 6 h for the starch to sediment. The supernatant was discarded, and the starch extract was dried overnight in an oven at 45 ± 5 °C.

### 2.2. Film Preparation

Edible films were prepared by dispersing Andean tube starches (5 g/100 mL) in the presence of glycerol (3 g/100 mL) into deionized water (MilliQ) under continuous stirring at 200 rpm and heating on a hot plate with a magnetic stirrer up to the starch gelatinization temperature (85–90 °C). After the heating process was completed, gels were degassed by applying a vacuum for 7 min. Then, the solutions were cooled to 50 °C, and approximately 40 mL of the film-forming solution was poured onto 100 × 10 mm glass plates (Petri dishes). Finally, the solutions were dried at 40 °C for 24 h. A similar procedure was used with the films, including mouthwash, substituting the deionized water with Listerine Original. All the casted samples were stored at approximately 58% RH and a temperature of 18 °C before characterization and sensorial analysis.

### 2.3. Optical Microscopy Characterization

Visual appearance and apparent transparency were determined using a conventional camera by reflecting diffused white light on the films placed onto a background containing black letters and figures over a white surface.

Optical micrographs were acquired in reflection mode with the help of an EVOS XL microscope and a 40×, 0.65 Numerical Aperture collimated objective using transmitted white illumination. Pieces of samples from each Andean tuber were carefully adhered to a microscope slide and placed inverted (upside down).

Color characterization was determined by means of a Lovibond LC 100 colorimeter, measuring at three distinct positions on each film.

### 2.4. Moisture Content

The moisture content (MC) of the films was determined gravimetrically. Samples of films extracted from each Andean tuber studied were separated from the original layer without any additional treatment and weighed ($W_{0}$). Pieces were dried at 105 °C for 24 h and weighed again afterwards ($W_{F}$). Measurement in triplicate of the MC was calculated using the equation
(1)$$\mathrm{MC}=\frac{W_{0}-W_{F}}{W_{0}}\;*100\;\%$$

### 2.5. Solubility

Total soluble matter (TSM) was determined on samples from each Andean tuber film studied. Squared 5 mm side pieces of each film were weighed ($W_{1}$), immersed into 30 mL of milliQ water, and left in constant and mild agitation under ambient conditions for 24 h. After recovering the remaining undissolved film portions by filtration, samples were dried at 105 °C for 24 h. Next, residues were weighed again ($W_{2}$). Values of the TSM were obtained in triplicate using the equation
(2)$$\mathrm{TSM}=\frac{W_{1}-W_{2}}{W_{1}}\;*100\;\%$$

### 2.6. Water Vapor Permeability

Water vapor permeability (WVP) was determined according to the method described by Sobral et al. [19]. Samples of films extracted from each Andean tuber were situated on the top part of a desiccator, above a milliQ water deposit existing at the bottom. For 10 h, all the pieces were exposed to saturated water vapor under ambient conditions. Afterwards, samples were situated on the top of a flask containing dried silica gel inside. Films blocked a circular 16.5 mm diameter aperture unique water vapor entrance to the flask interior. All chambers were positioned inside the same water vapor-saturated desiccator described above and weighed every hour for 10 h. Linear regression of the weight (w) values versus time (t) allowed obtaining the w/t data of the following WVP equation, which permitted calculating its value. Measurements were done in triplicate.
(3)$$\mathrm{WVP}=\frac{w}{tA\;}\frac{x}{∆P}$$

Thickness (x) was measured using a 0.01 mm conventional caliper and averaging the values obtained at 8 different locations of each film. The thickness measured was around 0.3 mm in all cases analyzed. The permeation area (A) was determined from the aperture of the flask. ∆P is the partial vapor pressure difference between the atmosphere with silica gel and pure water (2642 Pa at 22 °C).

### 2.7. Optical Absorption

A UV-visible light spectrophotometer was used to determine the opacity of each film analyzed. Samples were cut in a rectangular shape and placed directly into quartz cells using an accuSkan GO UV-Vis spectrophotometer (Thermo Fisher Scientific, Waltham, MA, USA). A blank quartz cell served as the reference value. Absorbance was recorded at 560 nm, and opacity (O) was calculated using the film thickness (x, mm) through the equation
(4)$$O=\frac{Abs_{560}}{x}$$

Measurements were performed in triplicate at ambient conditions.

Fourier transform infrared (FTIR) spectra were recorded according to the method described by Orsuwan et al. [20] using a Spectrum Two spectrophotometer (Perkin-Elmer, Waltham, MA, USA) coupled to an attenuated total reflectance (ATR) adapter. All measurements covered a wave number range between 4000 and 600 cm^−1^ with 4 cm^−1^ accuracy. Under ambient conditions, samples of each film were located directly on the ATR tip surface and gently pressed with the flat-tipped plunger. Data were acquired in triplicate at different locations of each film.

### 2.8. Elastic Properties

The mechanical properties of edible films, such as tensile strength (TS), elongation at break (EB), and elastic modulus (EM), were measured under ambient conditions by using the standard ASTM D 882–88 method (Instron Universal Testing). Films were cut into rectangular strips of 3 cm × 10 cm and preconditioned at 50% RH at ambient temperature for 48 h. The crosshead speed was set at 12.5 mm/min, and at least 5 replicates of each specimen were averaged together.

### 2.9. Atomic Force Microscopy

The surface morphology of the films extracted from each Andean tuber analyzed was characterized at the nanoscale using a Park Systems XE7 Atomic Force Microscope (Santa Clara, CA, USA). Samples were cut into small pieces and attached to the AFM sample holder with double-sided tape. Without any additional treatment, film surfaces were scanned in non-contact mode (*tapping*) under ambient conditions using PPP-NCHR commercial silicon cantilever tips (42 N/m, 330 kHz, <10 nm typical diameter). All images were acquired at 512 × 512 pixels^2^. AFM image edition was restricted to only a single polynomial leveling, performed using the Park System’s XEI software.

### 2.10. Statistical Analysis

Statistical analysis was performed through the determination of variance (ANOVA) and Tukey tests of multiple comparisons with the aid of the Statistic GraphPad Prism 8 software (GraphPad Software, San Diego, CA, USA). The significance level was set at 5%.

### 2.11. Sensorial Analysis

Twenty-seven judges evaluated several sensorial characteristics of films containing mouthwash by each tasting a piece extracted from each Andean tuber analyzed. In a single presentation, the following factors were graded: acceptability, texture, taste, odor, and utility as a mouthwash in this presentation (applicability). For all the features except texture, the panelists used a 5-point hedonic scale (5: very pleasant; 4: pleasant; 3: neither pleasant nor unpleasant; 2: unpleasant; 1: very unpleasant). For texture quality, the following scale was used: 5—very hard; 4—hard; 3—neither hard nor soft; 2—soft; 1—very soft.

## 3. Results and Discussion

Determining the properties of biofilms based on starch extractions from tubers is necessary to optimize the selection of the source, depending on the purpose [5,8,9]. An identical procedure of starch extraction was applied to the four Andean tubers analyzed: melloco, mashua, oca, and potato. Their macroscopic and nanoscopic properties are compared.

### 3.1. Macroscopic Optical Differences

Slight differences in the visual appearance and apparent transparency can be detected even in macroscopic observations, as shown in Figure 1 and in Appendix A.

At the microscopic scale, it is possible to distinguish more differences among them, as depicted in Figure 2 and in Appendix A. Since white light completely crosses the film thickness, dissimilarities among them may indicate the existence of internal regions of different compositions.

In order to quantify the color and opacity macroscopic variability, measurements of the chromaticity parameters L*, a*, and b*, as well as whiteness index (WI) and opacity data, were obtained from each Andean tuber studied. According to the CIELab scale, L* represents the sample degree of lightness, a* represents the redness, and b* represents the yellowness [21]. The instrument was calibrated against a standard white reference plate. Results are shown in Table 1.

Lightness was high in all films but even higher in the films extracted from melloco (*p* ≤ 0.05). Parameters a* and b* also showed variability (*p* < 0.05). Considering the L*, a*, and b* values as a whole, all films display a tendency to whitish tones, a common characteristic of starch-based films [10,22]. In regard to their opacity, recorded data are between 0.74 (oca) and 2.03 (potato), which was lower than those observed by Santacruz et al. [23], whilst mashua (1.29) and melloco (1.52) presented intermediate values, the last one in agreement to the data reported by Daza et al. [24].

### 3.2. Water Interaction

Since many potential applications of starch-based films are related to interactions with water, moisture content (MC), water activity (WA), total soluble matter (TSM), and water vapor permeability (WVP) behavior of samples of films extracted from each Andean tuber analyzed were compared. Results are shown in Table 2.

The contrast between the biofilms analyzed persists when studying their interaction with water. The lowest MC was found on the biofilms extracted from potato, and no significant differences were observed between the MC values of blends from melloco, mashua, and oca. The WVP data obtained for films yielded from melloco and mashua were higher than those related to oca and potato. This could be attributed to a strong interaction of starch-starch molecules in potato and oca films and a more flexible starch chain in melloco and mashua films [3]. Solubility could be related to the hydrophilic nature of starch-based films: those with greater crystalline starch content are sensitive to moisture and relative atmospheric humidity [25,26].

### 3.3. Fourier Transform Infrared Absorption

Structural properties may occur due to molecular bonding divergences, which can be inspected by exploring the absorption of infrared light frequencies of the biofilms using an attenuated total reflectance (ATR) adapter coupled to a Fourier transform infrared spectrophotometer. Data obtained are depicted in Figure 3.

Structural analysis performed by infrared absorption suggests a very close conformation among all the biofilms analyzed. All the spectra found present a broad band at around 3350 cm^−1^, corresponding to the free hydroxyl groups (-O-H), which are associated with the starch and glycerol components of the blends. The intensity of this broad band of the Andean tuber starch-based films could be related to the number of hydroxyl groups [27]. Sharp bands at 2933 and 2883 cm^−1^ are characteristic of C-H stretch, occurring at the starch [28]. Bands at 1633, 1452, 1404, and 1240 cm^−1^ can be assigned to the bending of water and CH_2_. The absorption peak at 1548 cm^−1^ could be related to the stretching of the C-H bond, and the band at 1337 cm^−1^ could be related to the C-O-H bending vibration [29], confirming the hydration capacity of the blends [30,31]. The bands in the region between 1200 and 900 cm^−1^ could be attributed to C-O, C-C, and C-O-H vibrations [28].

### 3.4. Elastic Properties

Mechanical properties measured on each film analyzed are plotted in Figure 4.

Films prepared from potato have the highest values of tensile strength (TS) and elastic modulus (EM) (21.6 and 1305 MPa, respectively), in agreement with the published data [9,32]. There is insufficient information in the literature regarding the mechanical properties of the Andean tubers studied in this work. However, the recorded values measured on films obtained from oca and potato are higher than those reported [33,34,35,36], proving the dependency of the blends on the extracting sources and on the yielding process. Nevertheless, starch-based films obtained from melloco tubers present similar mechanical properties to those reported [35]. Values of elongation at break (EB) of starch films are not in direct relationship to TS and EM, since the highest EB measurement corresponds to oca.

### 3.5. Nanoscopic Characterization

#### 3.5.1. Topography

Atomic force microscopy (AFM) permits detecting surface lateral irregularities of less than a few nanometers and sub-nano metrical vertical height contrasts. Avoidance of any coating requirement entails a significant biological potential applicability. In order to minimize lateral forces during the measurement, samples were scanned on *tapping* mode. A comparison among surface corrugations at different range spans of biofilms extracted from each different Andean tuber analyzed is shown in Figure 5.

AFM characterization is not constrained to a bidimensional surface morphology picture at the nanoscale. Recorded files comprise a set of data that allows, for instance, to draw tridimensional charts, as depicted in Figure 6. Appendix A show 3D images at other scan ranges.

The granular but smooth topography observed in the case of films extracted from melloco may explain the elastic properties observed. The existence of inhomogeneities at this scale underneath the surface can cause the appearance of defects that can be enlarged under increasing stress, weakening their whole structure. In the mashua and oca cases, the fibrous appearance shown in the 2 × 2 µm^2^ pictures reveals a similar effect. Potato starch-based films expose a more rigid surface at the nanoscale, both in topography and in phase contrast, indicating a harder structure with relatively less capability to adhere water molecules to their surface, in agreement with EM and TSM values measured. Interestingly, optical microscopy images of both melloco and potato cases suggest internal component separation, whilst films extracted from mashua and oca were apparently more homogeneous. The fibrous structure of these last samples was only discovered after AFM nanoscopic characterization.

#### 3.5.2. Phase Contrast

By using *tapping* mode, phase contrast can be recorded, which is related to the different energy dissipation occurring at every pixel of the image. These data provide information about elasticity properties or the tip-sample adhesion at the nanoscale [16]. Phase contrasts found at the nanoscale between the biofilms analyzed evidenced structural dissimilarities, as shown in Figure 7.

Phase contrast segregation observed could be attributed to a polarity change at the interface. Higher polarity regions may enhance the adhesion of ambient water to the surface, triggering an extra tip-sample grip by the emergent meniscus. Hence, phase contrast segregation, forming clear micron order structures in these conditions, suggests a water-dissolving capacity. This is found in the films yielded from melloco, mashua, and oca tubers, but it is more significant in the case of melloco. This result correlates to the TSM data measured.

### 3.6. Sensorial Analysis

The sensorial evaluation of starch-based biofilms containing mouthwash is summarized in Figure 8.

Films obtained from melloco tuber were rated as “*pleasant*” in all the parameters analyzed (odor, taste, applicability, and acceptability), indicating that the panelists accepted this sort of film well. On the other hand, the biofilms extracted from mashua, oca, and potato were qualified as “*neither pleasant nor unpleasant*” in all parameters, reflecting similarities among them. Regarding the texture feature, blends obtained from mashua, oca, and potato were rated as “*neither hard nor soft*”, but the biofilms produced from melloco were assessed as “*soft*”.

Findings at the nanoscale indicate that the use of starch films extracted from melloco tuber might be appropriate to perform orally disintegrated blends. The combination of a granular but smooth surface to a micron-order phase contrast segregation suggests a moderately rigid sample in dry conditions with a propensity to be dissolved in water. Therefore, a matrix based on starch extracted from melloco can maintain its shape in dry conditions, but after being bitten, films can be dissolved completely. This feature is appealing for orally disintegrated blends. The sensorial analysis confirmed the hypothesis.

## 4. Conclusions

Starch-based biofilms have different properties depending on the source from which they were extracted. In this work, films developed from the Andean tubers melloco, mashua, oca, and potato are analyzed. Macroscopic properties such as optical contrast, colorimetry, opacity, moisture content, water activity, total soluble matter, water vapor permeability, infrared optical absorption, and elastic properties are compared. Nevertheless, significant differences are found at the nanoscale, both in topography and phase contrast. These differences correlate to the macroscopic data observed, shedding light on the understanding of the causes originating the variation observed.

We discovered that melloco blended films present with high solubility, low tensile strength, and significant surface phase contrast segregation. Furthermore, their topography at the nanoscale shows a granular but compact structure. Hence, we propose these starch-based films as a potential oral disintegrating starch matrix. We include mouthwash on layers extracted from each Andean tuber studied here and performed a sensorial analysis. As expected, melloco samples obtained the greatest acceptability.

## Figures and Tables

**Figure 1 polymers-14-04116-f001:**

Visual appearance and apparent transparency of starch films extracted from each Andean tuber analyzed.

**Figure 2 polymers-14-04116-f002:**
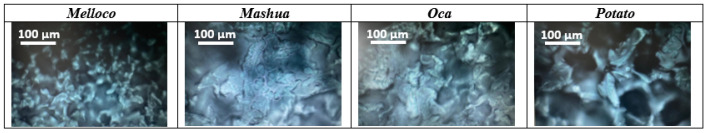
Micrographs of starch films extracted from each Andean tuber analyzed in reflection using white light transmission illumination.

**Figure 3 polymers-14-04116-f003:**
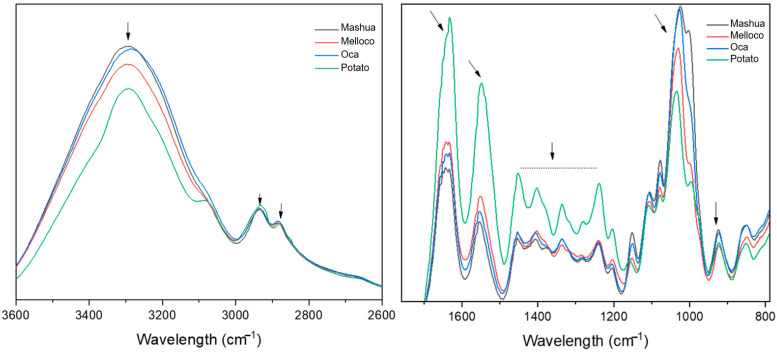
Fourier transform infrared absorption spectra measured on the films formulated with starch from the Andean tubers studied. Arrows highlight representative peaks.

**Figure 4 polymers-14-04116-f004:**
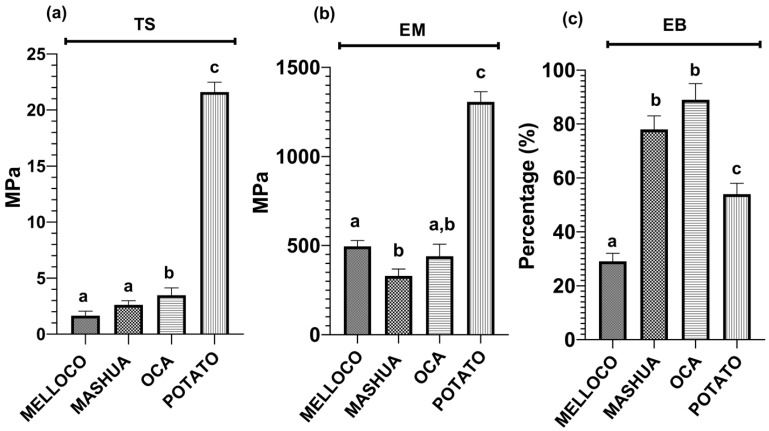
Mechanical properties obtained on the films extracted from each Andean tuber analyzed. Data shown are representative of five independent experiments and represent the mean ± SD. Different letters above the bars indicate significant differences (*p* < 0.05 ANOVA post hoc Tukey’s test). (**a**) Tensile strength, (**b**) elastic modulus (Young’s modulus), and (**c**) elongation at break.

**Figure 5 polymers-14-04116-f005:**
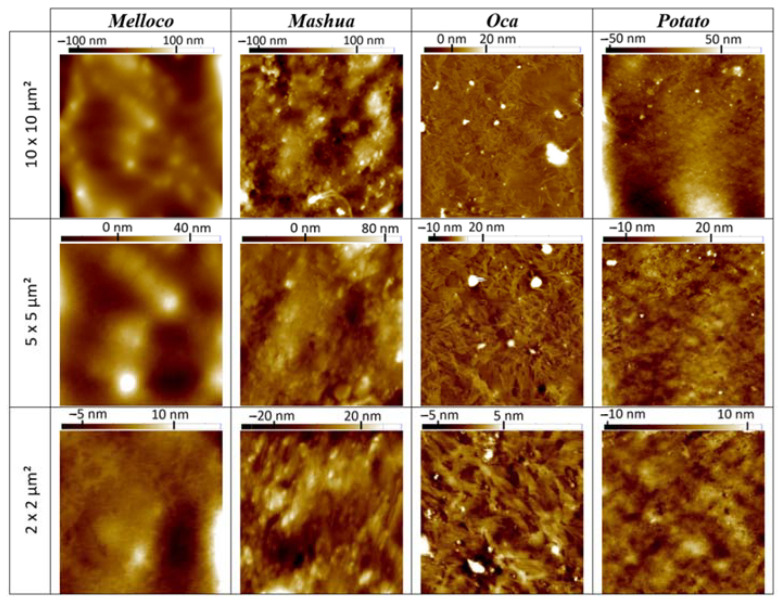
Topographic atomic force microscopy images measured using *tapping* mode on the surface of biofilms extracted from each Andean tuber studied at different scan area ranges.

**Figure 6 polymers-14-04116-f006:**
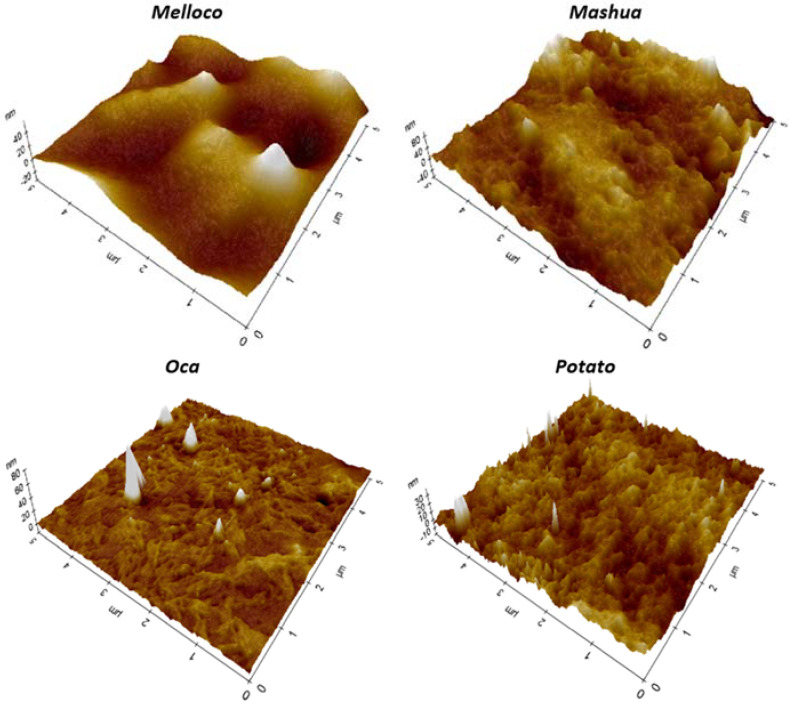
Tridimensional topographic atomic force microscopy comparison among 5 µm side squared images of biofilms extracted from each Andean tuber analyzed.

**Figure 7 polymers-14-04116-f007:**
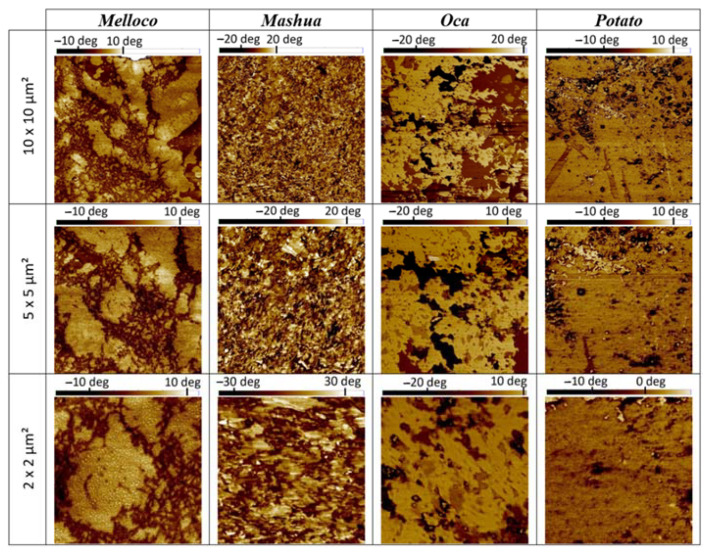
Phase contrast (*tapping* mode) images measured on films extracted from each Andean tuber analyzed at different spans.

**Figure 8 polymers-14-04116-f008:**
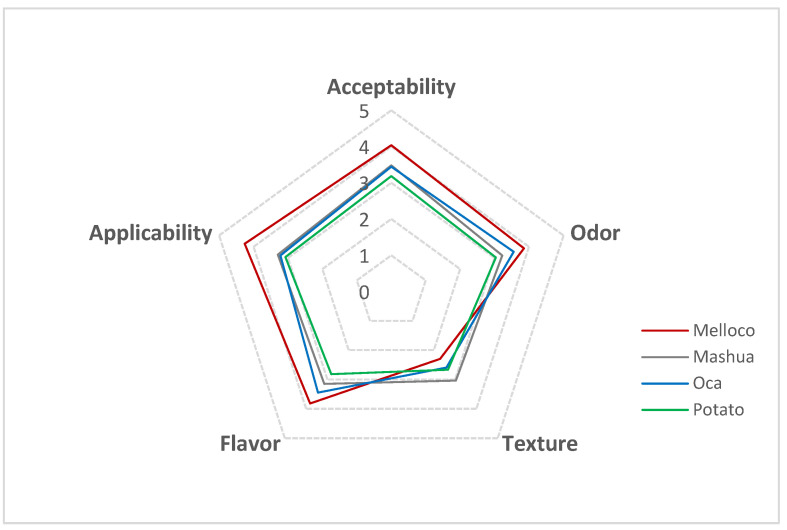
Sensorial evaluation of films extracted from each Andean tuber analyzed, blended with mouthwash.

**Table 1 polymers-14-04116-t001:** L*, a*, and b* color parameters, whiteness index (WI) and opacity data measured on films yielded from each Andean tuber analyzed.

Properties	Melloco	Mashua	Oca	Potato
L*	90.36 ± 0.25 ^a^	86.30 ± 0.82 ^b^	80.73 ± 1.09 ^c^	87.03 ± 0.98 ^abc^
a*	−0.27 ± 0.05 ^b^	0.47 ± 0.06 ^a^	0.63 ± 0.05 ^a^	0.40 ± 0.01 ^a^
b*	9.18 ± 0.15 ^b^	6.37 ± 0.06 ^c^	10.00 ± 0.26 ^ab^	11.40 ± 0.17 ^a^
WI	86.70 ± 0.19 ^a^	84.88 ± 0.72 ^ab^	78.28 ± 1.09 ^c^	82.72 ± 0.71 ^bc^
Opacity	1.52 ± 0.04 ^b^	1.29 ± 0.04 ^c^	0.74 ± 0.03 ^d^	2.03 ± 0.05 ^a^

Results are mean ± Standard Deviation (SD). Different letters (a, b, c, d) in the same row indicate significant differences between the different films (*p* ≤ 0.05).

**Table 2 polymers-14-04116-t002:** Moisture content (MC), water activity (WA), total soluble matter (TSM), and the water vapor permeability (WVP) data comparison among the starch-based films blended from each Andean tuber studied.

Properties	Melloco	Mashua	Oca	Potato
MC (%)	17.25 ± 0.25 ^a^	16.73 ± 0.15 ^a^	17.24 ± 0.18 ^a^	13.76 ± 0.22 ^b^
WA (a_w_)	0.5060 ± 0.004 ^c^	0.5218 ± 0.003 ^b^	0.5082 ± 0.003 ^a^	0.5079 ± 0.004 ^d^
TSM (%)	84.45 ± 0.12 ^a^	63.21 ± 0.10 ^c^	63.71 ± 0.11 ^b^	24.43 ± 0.13 ^d^
WVP (g mm min^−1^ m^−2^ kPa^−1^)	0.081 ± 0.001 ^bcd^	0.085 ± 0.001 ^ab^	0.054 ± 0.003 ^d^	0.058 ± 0.002 ^c^

Results are mean ± SD. Different letters (a, b, c, d) in the same row indicate significant differences between the different films (*p* ≤ 0.05).

## Data Availability

Not applicable.

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
