# Peer review of "Nanoscopic Characterization of Starch Biofilms Extracted from the Andean Tubers Ullucus tuberosus, Tropaeolum tuberosum, Oxalis tuberosa, and Solanum tuberosum"

_polymers, 2022, doi:10.3390/polym14194116_

Round 1
Reviewer 1 Report
Section - Introduction: (Line 57 – 64)
“Need to add a highlighting sentence novelty”
This research is quite good for the exploration stage of new starch materials for biofilm applications. However, the novelty in this study is less visible. This needs to be highlighted in the introduction at the end of the paragraph. For example: source, yield, and availability in the Ecuador area which may differ from starch that already exists and has been studied by several previous researchers.
In Materials & Methods.
Need improvement!
It is necessary to explain the amylose and amylopectin content of each type of starch. Because, the starch content will affect the properties of moisture, solubility and also mechanical properties.
Line 74 : Then, filtered using a cheesecloth.
what is the mesh size (cheesecloth)?
Line 79 : under continuous stirring
How many RPM rotation during stirring process?
Section 2.2 Film preparation
After drying the edible film in the oven at 40 oC for 24 hours, is the edible film directly tested? or stored under certain conditions (RH and temperature)?
How many repetitions of each variation in the test, especially moisture content, solubility, Water vapor permeability? Give and add your explanation in section 2.4 – 2.6.
Line 181-182
In table 1, there is an asterisk (*). Please provide a description for this marker.
Line 290 – 291
“from potato have the highest values of TS and EM (21.6 and 1305 MPa, respectively), which could be related to their amylose content.
Is there no test for the content of amylose and amylopectin in each type of tuber in your research?
amylose and amylopectin greatly affect the mechanical properties of the biofilm.
Author Response
Response to Reviewer 1 comments
- Section - Introduction: (Line 57 – 64)
“Need to add a highlighting sentence novelty”
This research is quite good for the exploration stage of new starch materials for biofilm applications. However, the novelty in this study is less visible. This needs to be highlighted in the introduction at the end of the paragraph. For example: source, yield, and availability in the Ecuador area which may differ from starch that already exists and has been studied by several previous researchers.
We acknowledge the comment of the reviewer. Indeed, we should have focused more on the importance this study has, not only by providing data on starch-based biofilms obtained from uncommon sources, but also for offering a simple way of detecting the viability of some films compared to others, depending on the application demanded. We have changed the paragraph by this one: “Here we show a comparative analysis of starch films extracted from melloco, mashua, oca, and potato (Solanum tuberosum L.). We measured at identical circumstances their optical contrast, colorimetry, opacity, Moisture Content (MC), Water Activity (WA), Total Soluble Matter (TSM), Water Vapor Permeability (WVP), infrared optical absorption, elastic properties, and compare the results to the nanoscopic surface characterizations using the AFM. Study provides a double objective: offering data from biofilms obtained from uncommon Ecuadorian starch sources, and stablishing a relationship between surface attributes at the nanoscale and macroscopic properties. Comparison proves the connection, showing that an AFM analysis on the surface of the starch-based biofilms is an easy and fast way to select the best kind of starch-based biofilm, depending on the specific demanding application. It is of quite importance in those cases where a complete chemical characterization is not straightforward.”
- In Materials & Methods.
Need improvement!
It is necessary to explain the amylose and amylopectin content of each type of starch. Because, the starch content will affect the properties of moisture, solubility and also mechanical properties.
We agree with the reviewer that the knowledge of the amylose and amylopectin content can shed more light on the comparison between the starch-based biofilms analyzed. However, in this work we wanted to provide a comparison between the macroscopic properties of several sort of films and their surface characteristics at the nanoscale. Hence, the amylose and amylopectin determinations should be done explicitly on the surfaces of the biofilms, instead than on the whole film. A study relating the macroscopic properties measured to the concentration of these components would be really interesting and will be considered in a future work. Nevertheless, it should be considered that fiber, fat, and other components might also play an important role, therefore it will require an exhaustive characterization. We consider that this is beyond the scope of the present work, for it focuses on the macroscopic properties and the nanoscopic AFM characterization comparison.
- Line 74 : Then, filtered using a cheesecloth.
what is the mesh size (cheesecloth)?
We forgot to state this parameter. We apologize for this error and thank the reviewer for noticing it. The next part has been added to the manuscript: “Then, filtered using a cheesecloth having a pore diameter of 120 mesh.”
- Line 79 : under continuous stirring
How many RPM rotation during stirring process?
We thank the reviewer for noticing that this value is missing. We have added this text to the manuscript: “…under continuous stirring at 200 rpm…”
- Section 2.2 Film preparation
After drying the edible film in the oven at 40 C for 24 hours, is the edible film directly tested? or stored under certain conditions (RH and temperature)?
It is true that films’ characteristics may alter due to media exposure, and we agree that this was missing. We thank the reviewer for letting us know and we have added the following text to the manuscript: “All the casted samples were stored at approximately 58% RH and 18 ºC temperature, before characterization and sensorial analysis”.
- How many repetitions of each variation in the test, especially moisture content, solubility, Water vapor permeability? Give and add your explanation in section 2.4 – 2.6.
Tests were repeated in triplicate. We thank the reviewer for warning us about this lack. It is described now in the sections 2.4-2.6.
- In table 1, there is an asterisk (*). Please provide a description for this marker.
Asterisk represents color parameters L*, a*, and b*, according to the CIELab scale. L* represents sample degree of lightness, whilst a* and b* are the sample chromaticity parameters. The instrument was calibrated against a standard white reference plate. In order to avoid potential confusion to the reader, we have clarified it in the text and a new reference has been added.
- “from potato have the highest values of TS and EM (21.6 and1305 MPa, respectively), which could be related to their amylose content. Is there no test for the content of amylose and amylopectin in each type of tuber in your research? amylose and amylopectin greatly affect the mechanical properties of the biofilm.
We agree that we should have described the data gathered for this work instead of assuming that the values measured occurred due to undetermined characteristics. We have modified the manuscript by substituting this part by the following: “…in agreement with the published data.”
Please see the attachment. It contains the modified manuscript, after having included the suggested changes.

Reviewer 2 Report
[1] The abstract should be improved, it does not convey any real information about the study to the reader (overly qualitative).
Introduction
[2] Line 27-29, this is an overstatement of the potential for starch. Please tone down this language.
[3] Line 37, not only tubers. There are a considerable range of vegetation that contain starch used for films (i.e. corn).
[4] Line 49-51, where is the evidence/citation?
[5] The aims are confusing. Are you aiming to show that AFM is a superior characterization technique? Or are you aiming to develop films to distribute mouthwash? Please clarify.
[6] There is no context in the introduction for the use of the mouthwash.
Materials and Methods
[7] Please give details of the tubers here, not in the introduction. Also provide quantities of the pulp and details of the mouthwash (% ethanol, sugars?, colorants? etc.).
[8] Section 2.2, how was the film contained on glass plates?
[9] Line 87, “Transparency and optical macroscopic appearance” is not the best description of this method. Rather “Visual appearance and apparent transparency” or similar.
[10] Changed “weighted” to “weighed” throughout.
[11] Line 137, what do you mean “layers”?
[12] There is no statement of statistical analysis.
Results
[13] I prefer a combined results and discussion section. It is clearer to the reader when results are discussed in the context of their presentation. Particularly in this case where results are presented separately with sectioning and the discussion section is one large block of text. It is difficult to follow the logic. It also avoids unnecessary repetition.
[14] Figure 4 is presented incorrectly. The x-axis of the left panel should be reversed.
[15] Figure 8 is confusing. Why not present 3 separate panels to avoid the splits in the y-axis of panel (a)?
[16] Table 4 is essentially repetition of Figure 8, choose one only.
Discussion
[17] Line 260, why is this observed? Please explain.
[18] Line 287-288, this is also highly dependent on the treatment of the data, i.e. normalization, baseline correction.
[19] Line 291, you cannot make this statement since you did not measure the amylose content.
[20] In general, the discussion is overly cursory with no in-depth evaluation of the differences observed. The discussion of the AFM is very weak, considering this is one of the aims. This sections needs significant improvement (see my comment about combined Results and Discussion).
Other comments
[21] I found the language difficult to follow. For example, lines 30-36 are not well written and need revision for clarity. I strongly recommend that the manuscript is fully revised by a competent English speaker or through an editing service.
Author Response
Response to Reviewer 2 comments
[1] The abstract should be improved, it does not convey any real information about the study to the reader (overly qualitative).
We thank the reviewer for noticing us this suggestion. The following sentences were added in order to improve the abstract: “Comparison permitted correlating macroscopic properties observed to the topography and tapping phase contrast segregation at the nanoscale. For instance, those samples presenting granular topography and disconnected phases at the nanoscale associate to less elastic strength and more water molecule affinity.”
[2] Line 27-29, this is an overstatement of the potential for starch. Please tone down this language.
Indeed, it was written too general and may confuse. We thank the reviewer for the observation reported. Sentence has been changed to this one: “Its green chemistry and reduced environmental impact make biopolymers a suitable alternative to traditional synthetic petroleum-based methods for many applications.”
[3] Line 37, not only tubers. There are a considerable range of vegetation that contain starch used for films (i.e. corn).
We agree that readers might confuse and think that we are stating that tubers are the only starch source. We have added a sentence and modified the previous one: “Tubers are a possible starch source. Although potato is widely used for this purpose, it is not the only tuber containing this carbohydrate.”
[4] Line 49-51, where is the evidence/citation?
Effect of reducing accuracy due to a metallic coating addition is something well known. However, since this comment could blur the message and it is not really necessary, we have decided to remove it. Instead, we have added another reference that shows the AFM capacity to measure biological samples.
[5] The aims are confusing. Are you aiming to show that AFM is a superior characterization technique? Or are you aiming to develop films to distribute mouthwash? Please clarify.
[6] There is no context in the introduction for the use of the mouthwash.
[5 and 6 reply] We thank the reviewer for pointing out that this part could not be very clear. We have changed the two paragraphs by these ones:
“Here we show a comparative analysis of starch films extracted from melloco, mashua, oca, and potato (Solanum tuberosum L.). We measured at identical circumstances their optical contrast, colorimetry, opacity, Moisture Content (MC), Water Activity (WA), Total Soluble Matter (TSM), Water Vapor Permeability (WVP), infrared optical absorption, elastic properties, and compare the results to the nanoscopic surface characterizations using the AFM. Study provides a double objective: offering data from biofilms obtained from uncommon Ecuadorian starch sources, and stablishing a relationship between surface attributes at the nanoscale and macroscopic properties. Comparison proves the connection, showing that an AFM analysis on the surface of the starch-based biofilms is an easy and fast way to select the best kind of starch-based biofilm, depending on the specific demanding application. It is of quite importance in those cases where a complete chemical characterization is not straightforward.
As an application utility, we explore the possibility of adding a mouthwash to these films and subject them to a tasting experiment. Melloco samples were selected as the best valued starch-based film for this purpose. Coincidentally, these samples display in dry conditions a consistent nanoscopic granular surface topography coupled to a clearer phase segregation contrast, which can be related to a better solubility and lower tensile strength. This proves the worth of an AFM inspection to optimize the biofilm selection, depending on the application required.”
[7] Please give details of the tubers here, not in the introduction. Also provide quantities of the pulp and details of the mouthwash (% ethanol, sugars?, colorants? etc.).
We have completed the Materials & Methods section by providing more information about the tubers. No other additives were included into the films, except those described in this section.
[8] Section 2.2, how was the film contained on glass plates?
We thank the reviewer for noticing us this error. We have substituted this part of the manuscript by the following text: “…were poured onto 100 x 10 mm glass plates (Petri dishes). ”
[9] Line 87, “Transparency and optical macroscopic appearance” is not the best description of this method. Rather “Visual appearance and apparent transparency” or similar.
We acknowledge the help. It has been changed accordingly.
[10] Changed “weighted” to “weighed” throughout.
It is now corrected.
[11] Line 137, what do you mean “layers”?
Part of the films. To state it clearer, this was substituted by “samples of each film”.
[12] There is no statement of statistical analysis.
Indeed, we agree with the observation. The following text has been added to the manuscript: “The statistical analysis was performed through an analysis of variance (ANOVA) and a Tukey test of multiple comparisons with the aid of the Statistic GraphPad Prism 8 software (GraphPad Software, San Diego California USA). The significance level was set at 5%”.
[13] I prefer a combined results and discussion section. It is clearer to the reader when results are discussed in the context of their presentation. Particularly in this case where results are presented separately with sectioning and the discussion section is one large block of text. It is difficult to follow the logic. It also avoids unnecessary repetition.
We intended to obey the Polymers’ template, which states: “3. Results. This section may be divided by subheadings. It should provide a concise and precise description of the experimental results, their interpretation, as well as the experimental conclusions that can be drawn. […] 4. Discussion. Authors should discuss the results and how they can be interpreted from the perspective of previous studies and of the working hypotheses. The findings and their implications should be discussed in the broadest context possible. Future research directions may also be highlighted.” Nevertheless, we have no problem at all merging both sections in one, as it appears now in the manuscript.
[14] Figure 4 is presented incorrectly. The x-axis of the left panel should be reversed.
We thank the reviewer for noticing this aspect. We have modified it accordingly. Picture has been corrected on the manuscript.
[15] Figure 8 is confusing. Why not present 3 separate panels to avoid the splits in the y-axis of panel (a)?
[16] Table 4 is essentially repetition of Figure 8, choose one only.
[15 and 16 reply] The manuscript is currently showing only the figure and we have removed the Table 4.
[17] Line 260, why is this observed? Please explain.
We have expanded the description of the optical microscopy images. However, ultimate causes of the differences found could be diverse. Nevertheless, observations agree with other data measured (i.e. AFM topographic picture). It is now stated in the text.
[18] Line 287-288, this is also highly dependent on the treatment of the data, i.e. normalization, baseline correction.
Considering that Figure 4 is representative enough, we have decided to omit this sentence.
[19] Line 291, you cannot make this statement since you did not measure the amylose content.
We agree that we should have described the data gathered for this work instead of assuming that the values measured occurred due to undetermined characteristics. We have modified the manuscript by just mentioning the following: “in agreement with the published data.”
[20] In general, the discussion is overly cursory with no in-depth evaluation of the differences observed. The discussion of the AFM is very weak, considering this is one of the aims. This sections needs significant improvement (see my comment about combined Results and Discussion).
Manuscript has been changed according to the comment regarding the Results and Discussion sections’ merger. Besides, text was considerably extended in this part, and now explains more details, particularly about the AFM results.
[21] I found the language difficult to follow. For example, lines 30-36 are not well written and need revision for clarity. I strongly recommend that the manuscript is fully revised by a competent English speaker or through an editing service.
We have changed the referred paragraph by this one: “Starch constitutes one of the best-known biofilm bases [2]. This polysaccharide can join numerous glucose molecules by glycosidic bonds, creating a structure that can be used as a multipurpose organic matrix. However, properties of these films are dependent on their chemical composition. Amylose/amylopectin percentage, plasticizer (as glycerol) concentration, or the presence of other trace elements at sample preparation, such as lipids and proteins, may entail important structural differences on each resulting starch film [3-8]. Hence, a proper selection of the starch source is of utmost importance to optimize the starch-based film attributes for a particular application.” Besides, we have simplified some sentences of the manuscript. We hope that now it is clear enough to understand the science involved in it.
Please see the attachment. It contains the modified manuscript, after having included the suggested changes.

Round 2
Reviewer 1 Report
Dear Authors,
Thank you for making some improvements to your article work.
Regards
Reviewers
Author Response
We would like to thank the efforts made on reviewing the manuscript entitled “Nanoscopic characterization of starch biofilms extracted from the Andean tubers Ullucus tuberosus, Tropaeolum tuberosum, Oxalis tuberosa, and Solanum tuberosum”, by Cynthia Pico, Jhomara De la Vega, Irvin Tubón, Mirari Arancibia, and Santiago Casado, to be considered for publication in Polymers as an article, in the special issue Advances in Bio-Based Polymeric Materials. Comments provided undoubtedly improved the manuscript.
We have corrected some minor typographic errors from the previous version.
Regards

Reviewer 2 Report
Thank you for your replies, the changes made are acceptable. In the future, please highlight the changes in the revision (i.e. different font color). This will be easier for reviewers to see the changes that have been implemented.
Author Response
We would like to thank the efforts made on reviewing the manuscript entitled “Nanoscopic characterization of starch biofilms extracted from the Andean tubers Ullucus tuberosus, Tropaeolum tuberosum, Oxalis tuberosa, and Solanum tuberosum”, by Cynthia Pico, Jhomara De la Vega, Irvin Tubón, Mirari Arancibia, and Santiago Casado, to be considered for publication in Polymers as an article, in the special issue Advances in Bio-Based Polymeric Materials. Comments provided undoubtedly improved the manuscript. Besides, we also thank the advice of highlighting the changes on the manuscript to make revision easier. We will proceed accordingly in the future.
We have corrected some minor typographic errors from the previous version.
Regards
